# GLOBAL-TO-LOCAL SUPPORT SPECTRUMS FOR MODEL EXPLAINABILITY

## ABSTRACT

Existing sample-based methods, like influence functions and representer points, measure the importance of a training point by approximating the effect of its removal from training. As such, they are skewed towards outliers and points that are very close to the decision boundaries. The explanations provided by these methods are often static and not specific enough for different test points. In this paper, we propose a method to generate an explanation in the form of support spectrums which are based on two main ideas: the support sets and a global-to-local importance measure. The support set is the set of training points, in the predicted class, that "lie in between" the test point and training points in the other classes. They indicate how well the test point can be distinguished from the points not in the predicted class. The global-to-local importance measure is obtained by decoupling existing methods into the global and local components which are then used to select the points in the support set. Using this method, we are able to generate explanations that are tailored to specific test points. In the experiments, we show the effectiveness of the method in image classification and text generation tasks.

## 1 INTRODUCTION

The ability to attribute a model's output to a particular set of training points or data source is important, especially in today's large generative models, given that most of the training data is scraped from the Internet. It gives us a way to explain the outputs of the models. It also supports, among other things, protection of copyright and intellectual property (i.e. the rights of data owners/creators), verification of reliability (e.g. of medical advices) and factuality (e.g. of news), and identification of problematic data points when the model behaves undesirably. The attributions not only assist users in understanding the reasoning behind the output, but also allow designers to "debug" the training data.

Several sample-based explanations have been proposed in the literature and two methods have risen to the forefront: influence functions and representer points. Influence functions measure the impact of removing a training point on the optimal parameters. It does so by computing the inverse of the Hessian which is then used to approximate the new optimal parameters by taking the first order Taylor approximation. When used to measure the influence of a training point on a test point, it can be seen as a gradient-based comparison method, since it computes the weighted dot product between the gradients of the two points. One limitation of the influence functions is their high computational cost associated with the Hessian. The representer points method, on the other hand, is based on the decomposition of the output of the neural network into contributions from the training points. The values of these contributions are then used as the measure of how important the points are.

However, we observe that these methods tend to be skewed by outliers, such as mislabled points or points that are very close to the decision boundaries (in the context of classification) or atypical input (in the context of generation), and therefore often attribute many different predictions or outputs to the same set of training points. In other words, these methods often produce global attributions of the model instead of local attributions that are more tailored to a particular prediction or output. While there are other methods that provide localized explanations, they often choose specific points and therefore lack the context of the full picture. In this paper, we propose a method with an adjustable global-to-local parameter that allow us to evaluate the shift in the data points from those that are

influential because they broadly affect the model to those that are particularly influential on a given test.

It shows a couple of advantages over existing methods: (1) They provide tailored explanations specific to the test points. Different test points within the same class may have spectrums with very different characteristics. We would like to point out here that the global important points are included in the spectrums as sub-explanations (they correspond to the points at the global extremes). (2) They show how well supported the test points are. Longer spectrums show a higher confidence in the model's prediction while shorter spectrums indicate the lack of support in the training set. These are useful indicators that can aid the process of data debugging and curation. Potential applications of the method in generative models include source attribution and identification problems in the generation process (like spurious correlation) and in the datasets (like the presence of biases). We show the effectiveness of the method in explaining predictions in classification problem, obtaining influential sources and detecting spurious correlations in generative language models which we view as an autoregressive (sequential) classification tasks.

## 2 SUPPORT SPECTRUMS

Many deep learning models consist of a feature map followed by a linear classifier. We will refer to the layer just before the linear classifier as the feature space. The main idea of our approach is to look at how the training points and the test point are distributed in the feature space. Consider a $T$-class classification problem with training points $Z = \{(x_1, y_1), \ldots, (x_n, y_n)\}$ where $x_i \in \mathbb{R}^d$ and $y_i$ is a one-hot vector of $T$ dimension specifying the class of $x_i$. We consider a neural network model which is defined by

$$\tilde{y}_i = \sigma(\Phi(x_i, \theta)) = \sigma(W f_i + b) \quad \text{where}$$
$$f_i = \Phi_2(x_i, \theta_2).$$

It consists of a feature map $\Phi_2$ with parameters $\theta_2$ which maps the input $x_i$ to its feature vector $f_i$ which is then passed to a linear classifier parameterized by weights $W$ and biases $b$. Collectively, the parameters of the whole network is denoted by $\theta = \{W, b, \theta_2\}$. $\sigma$ is a nonlinear function which transforms the output of the linear layer $\Phi(x_1, \theta)$ into the prediction vector. Note that there is no restriction placed on the feature map $\Phi_2$ which can be arbitrarily complex (deep). The model, therefore, encompasses a wide range of architectures including Transformer-based generative language models.

The goal of training is to find the optimal parameters that minimize $\frac{1}{n} \sum_{i=1}^{n} L(z_i, \theta)$ for some loss function $L$. Throughout this section, we will denote by $\hat{\theta} = \{\hat{W}, \hat{b}, \hat{\theta}_2\}$ the parameters obtained from training, which is an approximation to the optimal solution. We will also denote by $\hat{W}_k$ the row vector associated with the class $k$, and $\hat{b}_k$ its bias. In the following sections, we will refer to the hyperplane $\hat{W}_k \hat{f} + \hat{b}_k$ as the discriminant of $k$. We are interested in providing sample-based explanations for a test point $z_t = (x_t, y_t)$ whose predicted class is $c$. Note that, $c$ is not necessarily the same as what is given by $y_t$. In this paper, we propose an explanation in the form of a list of training points (a spectrum) that supports the classification. This list is obtained through a 2-step process: The generation of the support set, and the computation of the global-to-local important (influential) points.

### 2.1 SUPPORT SETS

Informally speaking, the support set is the set of training points of class $c$ that lie in between $x_t$ and the training points of the other classes in the feature space. They act as a "buffer" that supports the classification of the test input as $c$. Let $\hat{f}_t = \Phi_2(x_t, \hat{\theta}_2)$ and $\hat{f} = \Phi_2(x, \hat{\theta}_2)$. For a class $k \in [T]$, the support set of $z_t$ relative to $k$ is defined as

$$R(z_t; k) = \{(x, 1_c) \in Z \mid w_k \cdot (\hat{f}_t - \hat{f}) > 0\}$$

where

$$w_k = \begin{cases} \hat{W}_c & \text{if } k = c \\ \hat{W}_c - \hat{W}_k & \text{if } k \neq c. \end{cases}$$

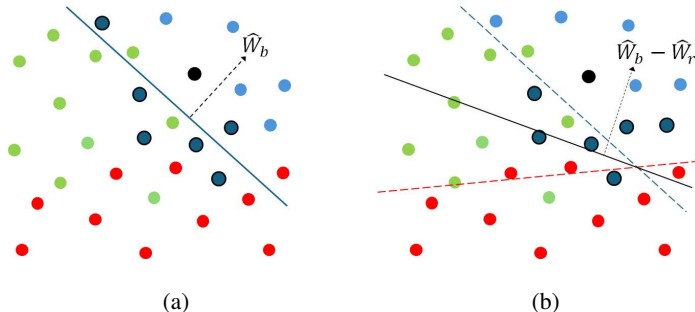

(a)            (b)

Figure 1: An illustrative example showing the training points (red, green, and blue dots) and a test point (black dot) in the learned feature space. (a) The blue line is the discriminant of the class blue whose normal is given by $\hat{W}_b$. The darker blue dots with dark edges are the general support set of the test point. (b) The solid black line is the decision boundary between blue and red, whose normal is $\hat{W}_b - \hat{W}_r$. The darker blue dots are the support set relative to red.

Here, we can distinguish between two types of support sets. We call $R(z_t; c)$, i.e. when $k = c$, the general support set and $R(z_t, k)$, when $k \neq c$, the support set relative to $k$. Figure 1 shows an illustrative example of both. The support sets give us an indication of how well supported the classification of $z_t$ is. When the support set is large it tells us that it is easier for the model to distinguish $z_t$ from the other classes. $R(z_t, c)$ indicates the overall distinguishability while $R(z_t, k)$ indicates how well $z_t$ can be distinguished from points in class $k$.

## 2.2 GLOBAL-TO-LOCAL SPECTRUMS

Existing sample-based valuations, such as representer points and influence functions, can be decomposed into a global component $g(z_i)$ which measures the overall importance of $z_i$ and a local component $\ell(z_i, z_t)$ which measures the importance of $z_i$ specific to $z_t$. Using this decomposition, we define the (global-to-local) spectrum of $z_t$ relative to the class $k$ as the set $S(z_t; k) = \{z_\delta \,|\, -\infty < \delta < \infty\}$ where

$$z_\delta = \arg \max_{z \in Z} \; g(z) \tag{1}$$

$$\text{subject to} \quad z \in R(z_t; k), \tag{2}$$

$$\ell(z, z_t) > \delta. \tag{3}$$

Constraint 2 restricts the spectrum to training points in the support set while Constraint 3 restricts the points to be in the locality of $z_t$ with $\delta$ being the controlling parameter. While the support set can be used as a form of explanation, its size can be very large and thus impractical. The idea behind the spectrums is, therefore, to select the most important points in the set.

As an example, consider a 3-class (red, green, and blue) classification problem as shown in Figure 2. In (a), the training points, together with the regions and decision boundaries of a trained network, are shown. In the rest of the figure, we show the important points as determined by different methods for two different test points (shown as black dots). In (d)-(e), the blue training points are shown according to their representer points values, where darker colors indicate higher values. For the two different test points, although there is a difference in emphasis, the top most important points are the same. In both cases, the blue dot that lies in the red region in the most important one. This illustrates how the valuation is skewed towards outliers, those that are close to the decision boundaries and those that are misclassified. Similar observations can be seen in (f)-(g) where the influence function is used. In contrast, (b)-(c) show the general spectrums (as points along the paths) computed using our approach. Here, the opaque blue dots are the support set. The two spectrums have the same starting point, but quickly move to important points in the locality of the test points, giving a better specialized explanation to each test point. Additionally, the spectrums can be tailored further to show how the test point is distinguished from specific classes as shown in Figure 3.

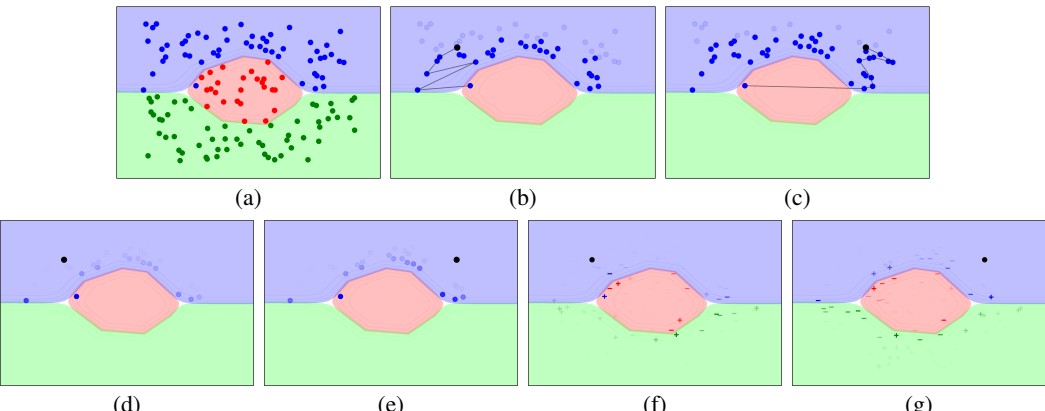

Figure 2: (a) The training points of a 3-class classification problem together with the regions and decision boundaries given by a trained network. (b)-(c) The general spectrums for two different test points (the black dots). Opaque dots indicate points that are in the support set, and transparent dots indicate those that are not. (d)-(e) Importance values given by the excitatory representer points. Darker colors indicate higher values. (f)-(g) Importance values given by the influence function. Similarly, darker colors indicate higher values.

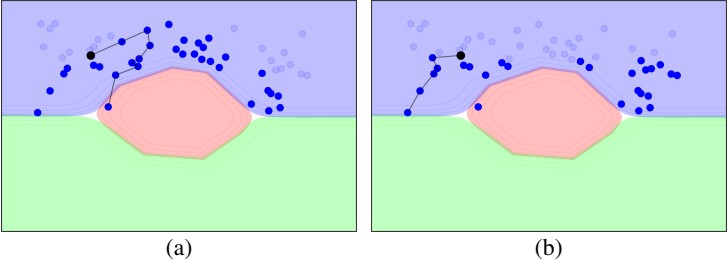

Figure 3: Spectrums for a test point (classified as blue) relative to (a) the red class and (b) the blue class. The opaque blue dots are the relative supports.

In the following, we show how the global and local importance values can be obtained from representer points and influence functions. Although we mainly focus on the representer points due to its efficiency, we describe both to show how the approach can be used with the latter.

## 2.3 REPRESENTER POINTS

The representer points method (Yeh et al., 2018; Schölkopf et al., 2001) is based on the result that if $\hat{\theta}$ is a stationary point of the minimization problem

$$\min_{\theta} \frac{1}{n} \sum_{i=1}^{n} L(z_i, \theta) + \lambda \|\theta\|_2^2 \quad \text{with} \quad \lambda > 0,$$

then the output of the linear layer on the test input $x_t$ can be written as

$$\Phi(x_t, \hat{\theta}) = \sum_{i=1}^{n} \frac{1}{-2\lambda n} \left[ \frac{\partial L(z_i, \hat{\theta})}{\partial \Phi(x_i, \hat{\theta})} \right] \hat{f}_i^\top \hat{f}_t. \tag{4}$$

In this work, we restrict $\sigma$ to be the softmax function and $L$ cross entropy loss which is the standard in classification tasks and so

$$L(z_i, \hat{\theta}) = -\sum_{j=1}^{T} y_{ij} \log \left( \frac{\exp(\bar{y}_{ij})}{\sum_{k=1}^{T} \exp(\bar{y}_{ik})} \right).$$

Substituting the above into Equation 4, we get

$$\Phi(x_t, \hat{\theta}) = \sum_{i=1}^{n} \frac{1}{2\lambda n}(y_i - \hat{y}_i)\hat{f}_i^\top \hat{f}_t,$$

where $\hat{y}_i = \sigma(\Phi(x_i, \hat{\theta}))$. From here, we can take the global and local importance of $z_i$ as $\left[\frac{1}{2\lambda n}(y_i - \hat{y}_i)\right]_c$ and $\hat{f}_i^\top \hat{f}_t$ respectively. Note that in order to obtain a maximizer of Problem 1, we don't need the exact values of the global importance as long as they give the same ordering. It can therefore be simplified to an equivalent expression given below. This is due to the fact that (1) softmax preserves ordering and (2) we are only concerned with the training points in $c$. In the end, we obtain the following simple functions:

$$g(z_i) = -(\hat{W}_c\,\hat{f}_i + \hat{b}_c) \quad \text{and} \quad \ell(z_i, z_t) = \hat{f}_i^\top \hat{f}_t,$$

where the global importance has the geometrical interpretation that the highest valued point is the one furthest in the opposite direction to the normal of the discriminant of the class $c$. The function $g$ above can be used to generate both the general and relative spectrums. However, empirical results show that using $\hat{W}_c - \hat{W}_k$ and $\hat{b}_c - \hat{b}_k$ instead of $\hat{W}_c$ and $\hat{b}_c$ yields a better spectrum relative to $k$.

## 2.4 INFLUENCE FUNCTIONS

Influence functions (Koh & Liang, 2017) approximate the changes in the optimal parameters due to perturbations in the loss contribution of a single training point. If $\hat{\theta}_z(\varepsilon)$ is the solution to the optimization problem

$$\min_\theta \frac{1}{n}\sum_{i=1}^{n} L(z_i, \theta) + \varepsilon L(z, \theta)$$

for some $z \in Z$, then under some conditions (most notably, the strict convexity of the objective function), it can be shown (Cook & Weisberg, 1982) that the rate of change of $\hat{\theta}_z(\varepsilon)$ at the point $\varepsilon = 0$ is given by

$$\left.\frac{d\hat{\theta}_z(\varepsilon)}{d\varepsilon}\right|_{\varepsilon=0} = -H_{\hat{\theta}}^{-1}\nabla_\theta L(z, \hat{\theta}),$$

where $H_{\hat{\theta}} = \frac{1}{n}\sum_{i=1}^{n}\nabla_\theta^2 L(z_i, \hat{\theta})$ is the Hessian. Using this fact, we can then apply the first order Taylor approximation to estimate the value of $\hat{\theta}_z(\varepsilon)$ for any $\varepsilon$. In particular, setting it to $-1/n$ approximates the optimal parameters when $z$ is removed from the training set. Note that the accuracy increases as $n$ grows larger. We can therefore use the magnitude of the rate as the global importance and set $g(z) = \|-H_{\hat{\theta}}^{-1}\nabla_\theta L(z, \hat{\theta})\|$. Furthermore, using the chain rule, one can approximate the loss on a test point when $z$ is removed using the following (which can be used as the local importance):

$$\ell(z, z_t) = \left.\frac{\partial L(z_t, \hat{\theta}_z(\varepsilon))}{\partial \varepsilon}\right|_{\varepsilon=0} = -\nabla_\theta L(z_t, \hat{\theta})^\top H_{\hat{\theta}}^{-1}\nabla_\theta L(z, \hat{\theta}).$$

One drawback of influence functions is the high computational cost associated with computing the inverse of the Hessian. However, there are already existing works that try to provide efficient approximations, and comparison with spectrums generated using these methods will be our main priority in future works.

## 3 EXPERIMENTS

In this section, we demonstrate the advantages of the spectrums as a form of explanations, in an image classification task (MNIST) and a text generation task (GPT2-XL & Open-LLaMA-7B).

## 3.1 MNIST

In this experiment, a convolutional neural network is trained to recognize hand-written digits in the MNIST dataset. Using the trained network, we generate the spectrums for some test points, two

| $z_t$ | $S(z_t, k)$ | $k$ | $z_t$ | $S(z_t, k)$ | $k$ |
|---|---|---|---|---|---|
| | | 0 | | | 0 |
| | | 1 | | | 1 |
| | | 2 | | | 2 |
| | | 3 | | | 3 |
| | | 4 | | | 4 |
| | | 5 | | | 5 |
| | | 6 | | | 6 |
| | | 7 | | | 7 |
| | | 8 | | | 8 |
| | | 9 | | | 9 |

Figure 4: The spectrums for two test points (with the same predicted class but different styles) from MNIST.

of which are shown in Figure 4. More examples can be found in the Appendix. In the figure, two test points whose predicted class is the same ($c = 5$) are shown in the first column. In the second columns, the general spectrums (row 6) and the relative spectrums (other rows) are shown. They are shown from locally (left) important points to globally (right) important ones. Although the predicted class is the same, the two test points are written in fairly different styles. This fact is reflected by the spectrums, where in every row, the two spectrums start (from the right) with the same point and then move to points that are closer in style to the respective test points, providing specific "evidences" in support of the classification.

Notice also that for a spectrum relative to a class $k \neq 5$, the point at the right most position exhibits some features that are associated with the class $k$ (see the case for $k = 2, 3, 4$, and $6$). These are the points that lie close to the decision boundary between the two classes and are important for distinguishing between the two. In particular, for the first test point, we see that it has a short spectrum relative to the class 6. This indicates that it not as well supported against (distinguished from) the class. And, indeed, the way it's written is close to the number 6.

## 3.2 GPT2-XL AND OPEN-LLAMA-7B

In this experiment, we test our approach on two text generation models: GPT2-XL and Open-LLaMA-7B. In the former, we take the pre-trained GPT-2 (1.6B) model and fine tuned it using OpenWebText ($\sim$9B tokens from $\sim$8M documents) which is an open-source replication of the original dataset used by GPT-2. In the latter, we take the Open-LLaMA-7B model which has been pre-trained on the RedPajama dataset and run our method on the Wikipedia data subset ($\sim$24B tokens from $\sim$7M articles). To compute the spectrums of a generated text, we treat the text generation process as a sequence of (autoregressive) classification tasks. In other words, the selection of each next token is viewed as classifying the preceding text (the prompt together with the tokens generated so far) into one of the $T$ classes where $T$ is the vocabulary size ($\sim$50K for GPT2-XL and $\sim$32K for Open-LLaMA-7B). Given the sheer number of classes, it is impractical to generate the relative spectrums. In this experiment, we will restrict ourselves to general spectrums (simply referred to as spectrums henceforth). It is possible to compute a spectrum for each token in the generated text. However, since the text can be quite long, we choose only the ones with high TF-IDF scores. A training point for a particular token $c$ is a sequence in the dataset of length $p$ that ends with $c$. Here, we set $p$ to range from 2 to the length of the input tokens plus a buffer (set to 20 tokens).

Looking at the spectrums of different tokens in the same text reveals some information about how these tokens are chosen and, generally, we can categorize the tokens into two groups: ones that are well supported and ones that are not. We start by showing some examples of well-supported tokens from both GPT2-XL (Figure 5) and Open-LLaMA-7B (Figure 6). In these figures, we show the top locally important sequences in the dataset for two different tokens of the generated text. For longer spectrums, we refer the reader to the Appendix, where we include sequences in the middle and at the (global) end of the spectrums. In both examples, the important sequences and their source articles are highly relevant to the generated text, that is, the selection of the token is well supported by the

| | |
|---|---|
| The origin of life is a hot topic among scientists. In a paper published this week in Nature Communications, researchers from the University of California, San Diego, and the University of California, Los Angeles say they have discovered a new way of making the building blocks of RNA | The origin of life is a hot topic among scientists. In a paper published this week in Nature Communications, researchers from the University of California, San Diego, and the University of California, Los Angeles say they have discovered a new way of making the building blocks of RNA. The researchers say that this method could be a way forward for making the complex molecules |
| Scientists may have figured out the chemistry that sparked the beginning of life on Earth. 

 The new findings map out a series of simple, efficient chemical reactions that could have formed molecules of RNA 

 **Source Article:** "How RNA got started" by Solmaz Barazesh. *ScienceNews*, 2009. | remarkable than its mutation rate. "It's extremely high," said Irene Chen, a Harvard University systems biologist who studies the evolution of molecules 

 **Source Article:** "Organism sets mutation speed record, may explain life's origins" by Brandon Keim. *Wired*, 2009. |
| of life. 

 All known life makes at least some use of RNA as a genetic material, and as the "R" in RNA 

 **Source Article:** "Missing building block of life could be made on ice in space" by Joshua Sokol. *NewScientist*, 2016. | a molecule from which the simplest self-replicating structures are made. Until now, they couldn't explain how these ingredients might have formed. "It's like molecular choreography, where the molecules 

 **Source Article:** "Life's first spark re-created in the laboratory" by Brandon Keim. *Wired*, 2009. |
| theory were outlined by Prof Steven Benner at the Goldschmidt meeting in Florence, Italy. 

 Scientists have long wondered how atoms first came together to make up the three crucial molecular components of living organisms: RNA 

 **Source Article:** "Earth life may have come from Mars" by Simon Redfern. *BBC News*, 2013. | The RNA World hypothesis states that RNA must have served important genetic and catalytic roles in life's early stages. On the other hand, some authors stress that a complex and unstable molecule like RNA could not have existed on its own without a supporting metabolic network composed of simpler molecules 

 **Source Article:** "The Astrobiology Primer v2.0" by Shawn D. Domagal-Goldman et.al. *National Library of Medicine*, 2016. |
| on a popular theory for how life on Earth began about four billion years ago. 

 The study questions the "RNA world" hypothesis, a theory for how RNA 

 **Source Article:** "News from the primordial world" by The Scripps Research Institute. ScienceDaily, 2016. | notion of how a protein behaves — de novo proteins have a more disordered architecture. That makes them a bit floppy, allowing the protein to bind to a broader array of molecules 

 **Source Article:** "How new genes arise from junk DNA" by Emily Singer. *World Economic Forum*, 2015. |

Figure 5: An example from GPT2-XL. The first row shows parts of the same generated text (the underlined parts are the prompt). Each column shows the top 3 training sequences (together with the source articles) that are closest to the generated text in its general spectrum which is computed, respectively, using the tokens RNA and molecules as the output.

| | |
|---|---|
| The proton is the smallest particle of an atom. It is a positively charged particle. It is found in the nucleus | The proton is the smallest particle of an atom. It is a positively charged particle. It is found in the nucleus of an atom |
| Aston's early model of nuclear structure (prior to the discovery of the neutron) postulated that the electromagnetic fields of closely packed protons and electrons in the nucleus 

 **Source:** ∼/Mass_(mass_spectrometry) | Though useful as a predictive model, the resulting screening constants contain little chemical insight as a qualitative model of atomic structure. 

 Comparison with nuclear charge 

 Nuclear charge is the electric charge of a nucleus of an atom 

 **Source:** ∼/Effective_nuclear_charge |
| repelling the others, giving a net lower electrostatic interaction with the nucleus. One way of envisioning this effect is to imagine the 1s electron sitting on one side of the 26 protons in the nucleus 

 **Source:** ∼/Effective_nuclear_charge | separate them. Therefore, the closer quarks are to each other, the less the strong interaction (or color charge) is between them; when quarks are in extreme proximity, the nuclear force between them is so weak that they behave almost as free particles. This is the reason why the nucleus of an atom 

 **Source:** ∼/David_Gross |
| particle obeys Bose statistics. If the nitrogen nucleus had 21 particles, it should obey Fermi statistics, contrary to fact. Thus, Heitler and Herzberg concluded: "the electron in the nucleus 

 **Source:** ∼/Discovery_of_the_neutron | Orbital Angular Momentum 

 When quantum mechanics refers to the orbital angular momentum of an electron, it is generally referring to the spatial wave equation that represents the electron's motion around the nucleus of an atom 

 **Source:** ∼/Orbital_motion_(quantum) |

Figure 6: An example from Open-LLaMA-7B. The first row shows parts of the same generated text (the underlined parts are the prompt). Each column shows the top 3 training sequences (in red) from Wikipedia (together with the source articles) that are closest to the generated text in its general spectrum which is computed, respectively, using the tokens nucleus and atom as the output. The symbol ∼ stands for "https://en.wikipedia.org/wiki". The texts in black precede the sequences and are provided for context.

spectrum. This further demonstrates that the method can be potentially used for source attribution in generative models. This is important in the context of today's generative models where most datasets are scraped from the Internet which might include copyrighted materials. The ability to attribute an output to the sources in the training data is therefore crucial. This will also be useful when the generated text is problematic, e.g. when it contains untrue statements or biases towards certain groups of people. It can help us in identifying the problematic sources and allowing us to take the necessary steps to correct the biases in the datasets.

| |
|---|
| The origin of life is a hot topic among scientists. In a paper published this week in Nature Communications, researchers from the University of California |
| This approach stands in stark contrast to the views advanced by organizations that focus specifically on AGI such as the Future of Life Institute at MIT, the Future of Humanity Institute at the University of Oxford, and the Machine Intelligence Research Institute at the University of California |
| of This could mean that Homo sapiens evolved earlier than we first thought or humans and Neanderthals weren't the first ones to use stone-tipped spears. Either way, the implications are huge. This announcement comes from lead author Yonatan Sahle from the Human Evolution Center at the University of California |
| "Wanted: Adventurous woman to give birth to Neanderthal," ran a British tabloid headline, after Harvard scientist, George Church, suggested cloning a Neanderthal. This reconstruction of a female Neanderthal is in Asturias, Spain. Photograph by Joe McNally, National Geographic What Novak and his team at the University of California |
| But the STAR V3 3D-printed robot is able to scoot 15 feet per second and flatten itself to get under doors, calling to mind the iris-scanning robots from the movie Minority Report. Developed by a team of researchers at the University of California |
| "We are delighted that the LHC is now cold again," says Beate Heinemann, the deputy leader of the ATLAS experiment and a physicist with the University of California |

In the table above, using the same generated text as in Figure 5, the spectrum for the token "California" is given where the top 5 locally important sequences are shown (in red) together with some preceding tokens for added context. In contrast to the previous ones, this example shows a case when a token is not well supported by its spectrum. One can notice that these sequences do not have any relevancy to the topic of the generated text. We can suspect, therefore, that the token "California" is selected not based on the full context of the previous text but only on some parts of it. Here, by evaluating the spectrum, we can arrive at the conclusion that the selection of the token is based on spurious correlation.

| Prompt | Top 5 next tokens |
|---|---|
| In a paper published this week in Nature Communications, researchers from the University of | California (0.59) Washington (0.18) Illinois (0.09) Cambridge (0.07) Texas (0.07) |
| As silicon-based computer chips approach their physical limitations in the quest for faster and smaller designs, the search for alternative materials that remain functional at atomic scales is one of science's biggest challenges. In a paper published this week in Nature Communications, researchers from the University of | California (0.59) Illinois (0.16) Texas (0.09) Maryland (0.09) Michigan (0.08) |
| It takes more than a galaxy merger to make a black hole grow and new stars form: machine learning shows cold gas is needed too to initiate rapid growth – new research finds. In a paper published this week in Nature Communications, researchers from the University of | California (0.59) Washington (0.12) Cambridge (0.1) Chicago (0.09) Texas (0.09) |
| Researchers flip the switch at the nanoscale by applying light to induce bonding for single-molecule device switching. In a paper published this week in Nature Communications, researchers from the University of | California (0.54) Illinois (0.19) Washington (0.09) Maryland (0.09) Texas (0.09) |

This suspicion is reinforced by a small experiment shown in the table above, where we ask GPT2-XL to predict the next token for sequences that end with the string "researchers from the University of". In the sequences, the string is preceded by different texts with different topics. The second column shows the softmax values when restricted to the top 5 tokens. We can see that, although the inputs are very different, the model returns "California" as the highest valued token for all. This shows that the selection of the next token depends heavily on the last few tokens in the inputs. One possible reason for this bias is the presence of a large amount of sequences in the dataset that contain the string "researchers from the University of" or its variants, followed by the token "California".

| |
|---|
| Q: What is the difference between diffraction and refraction? |
| A: Refraction is the scattering of electromagnetic waves |

| |
|---|
| is based on the theories of Planck and the interpretation of those theories by Einstein. The correspondence principle then allows the identification of momentum and angular momentum (called spin), as well as energy, with the photon. Polarization of classical electromagnetic waves **Source Article:** ∼/Photon_polarization |
| Spin angular momentum may refer to: Spin angular momentum of light, a property of electromagnetic waves **Source Article:** ∼/Spin_angular_momentum_(disambiguation) |
| x 10-7 H/m. = relative magnetic permeability of the material. Magnetically conductive materials such as copper typically have a near 1. This velocity is the speed with which electromagnetic waves **Source Article:** ∼/Speed_of_electricity |

The same phenomenon can be observed for some of the examples generated by Open-LLaMA-7B. One is given in the table above. Here, one might suspect that the selection of the token "waves" depends heavily on the immediate preceding token "electromagnetic". Although in this case, more evaluations are needed to confirm this suspicion, especially on the rest of the RedPajama dataset. Looking at the examples above, one might suggest that the tokens are not well supported because of the fact that the generated texts are simply too short. In comparison, the input texts for the tokens "RNA" and "molecules", in the example of Figure 5, are relatively long. In this final example, we show that this is not necessarily the case. In the following table, the top 5 locally important sequences are given for the token "genetic". In this case, the input text is longer than previous examples, and the token has one of the highest TF-IDF scores, and yet the spectrum shows little relevancy to the generated text. The reason for this, however, is less clear than in the previous case. One possibility is that the selection is simply due to the presence of the tokens "DNA" and "RNA", without looking at the larger context of the input.

| |
|---|
| The origin of life is a hot topic among scientists. In a paper published this week in Nature Communications, researchers from the University of California, San Diego, and the University of California, Los Angeles say they have discovered a new way of making the building blocks of RNA. The researchers say that this method could be a way forward for making the complex molecules that are used to make DNA and other genetic |
| ) and other functional RNAs has shown how these seemingly simple molecules can carry out the complex functions of proteins. Jennifer Doudna: Gene drives are ways that a gene editor can be used to spread a genetic |
| bacteria, they are distinct. Archaea inhabit many harsh environments. bacterium (plural bacteria) A single-celled organism. These dwell nearly everywhere on Earth, from the bottom of the sea to inside animals. DNA (short for deoxyribonucleic acid) A long, double-stranded and spiral-shaped molecule inside most living cells that carries genetic |
| all cellular life forms. The actual translating from DNA/RNA language into protein language involves an RNA called transfer RNA, or tRNA for short. tRNA is the key to understanding the nature of the genetic |
| with knots in them. We study DNA because it serves as a good model polymer that's analogous to the much smaller polymer molecules that plastics and other modern materials are made of, and we study knots because we want to understand how entanglement affects polymer dynamics. There are also some genetic |
| which include bacteria, and eukaryotes, which include plants and animals. While other evolutionary biologists had long studied physical traits of species to determine their relationships, Dr. Woese spent years laboriously comparing the genetic |

In summary, through the experiments, we have shown how our approach provide a richer sample-based explanations. In contrast to existing methods that output individual training points that are often static, our approach output a spectrum of points that shows the transition from global to local importance. In the MNIST case, we show that they indicate how distinguishable the test point is relative to the different classes. They also give tailored explanations to different test points within the same class. They show that points in the same class might have different levels of distinguishability, which is not apparent when standard methods like influence functions and representer points are

used. In the case of text generation, we show how they can be used to (1) generate important sources for attribution, in the case when the tokens used are well supported, and (2) identify problems in the generation process (like spurious correlations) and in the dataset (like the presence of biases).

# 4 RELATED WORKS

Existing approaches for estimating the influence of a test point to a model's prediction include influence functions, representer points, TracIn and model retraining.

Influence functions (Koh & Liang, 2017; Hampel, 1974; Cook, 1977) are based on first-order Taylor approximations of the loss function under small perturbations to the model such as removing a single training point. Its applications include explaining predictions (Koh & Liang, 2017), producing confidence intervals (Schulam & Saria, 2019), investigating model bias (Brunet et al., 2019; Wang et al., 2019), and crafting data poisoning attacks (Koh et al., 2022). Barshan et al. (2020) gives a relative variant of the method. Representer points methods (Yeh et al., 2018) are based on representer theorems (Schölkopf et al., 2001; Bohn et al., 2019; Unser, 2019; 2021) that decompose a prediction of a network into a linear combination of contributions from the training points. Its main application is in providing post-hoc explanations for deep neural networks (Yeh et al., 2018; Sui et al., 2021). TracIn (Pruthi et al., 2020; Yeh et al., 2022) compares the trajectories of the training gradients; and model retraining (Ghorbani & Zou, 2019; Jia et al., 2019; Kwon & Zou, 2022; Feldman & Zhang, 2020) uses repeated trainings to derive importance measures based on game theoretic concepts like the Shapley value.

# 5 CONCLUSIONS

In this paper, we propose a method to generate a global-to-local support spectrums for explaining the output of a black-box model on a test point. It is based on the idea of support sets and the decoupling of existing sample-based methods into the global and local components. A point in the spectrum is obtained my maximizing the global importance subject to the locality constraint and the full spectrum is generated by varying the locality parameter. In the experiments, we show the effectiveness of the explanations in two example tasks: image classification and text generation. As future works, we would like to study how the generated spectrums differ when different global/local measures are used, to scale the approach to the datasets of state-of-the-art models, and to apply it to other tasks like image generation. Finally, we would like to explore applications of the method to related domains of active learning, data selection and valuation.

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

## A  ADDITIONAL MNIST EXAMPLES

## B ADDITIONAL LLAMA2-7B EXAMPLES

~ = https://en.wikipedia.org/wiki

The theory of relativity is a theory of space and time. It was developed by Albert Einstein

is how general relativity can be reconciled with the laws of quantum physics to produce a complete and self-consistent theory of quantum gravity.

From special to general relativity

In September 1905, Albert Einstein

**Source Article:** ~/Introduction_to_general_relativity

then expect light to be moving towards us with a speed that is offset by the speed of the distant emitter (c ± v).

In the 20th century, special relativity was created by Albert Einstein

**Source Article:** ~/Emission_theory_(relativity)

relative motion may experience an electric current and a magnetic field. Thus the one field (called the electromagnetic field) may appear in different guises.

In his famous 1905 'Special Relativity' paper, Albert Einstein

**Source Article:** ~/James_Clerk_Maxwell_Foundation

. . .

approach, Riemannian geometry was maintained, and the extra dimension allowed for the incorporation of the electromagnetic field vector into the geometry. Despite the relative mathematical elegance of this approach, in collaboration with Einstein and Einstein

an Westerveld) – The King of France.
 Queen Anne (portrayed by Rebecca Eady) – The queen consort of King Louis XIII.
 Albert Einstein

aja" Einstein (18 November 1881 – 25 June 1951) and her older brother, Albert, were the two children of Hermann Einstein

. . .

lunar impact crater that is located near the western limb of the Moon. It is named for the Portuguese explorer Vasco da Gama. It lies to the south of the walled plain Einstein

historical novels
Novels by Frank Yerby
Dial Press books
Novels set in the 19th century
Novels set in Mississippi Einstein

erences

External links

Colleges in Malaysia
Universities and colleges affiliated with the Seventh-day Adventist Church Nursing schools in Malaysia Einstein

---

The proton is the smallest particle of an atom. It is a positively charged particle. It is found in the nucleus

Aston's early model of nuclear structure (prior to the discovery of the neutron) postulated that the electromagnetic fields of closely packed protons and electrons in the nucleus

**Source Article:** ~/Mass_(mass_spectrometry)

repelling the others, giving a net lower electrostatic interaction with the nucleus. One way of envisioning this effect is to imagine the 1s electron sitting on one side of the 26 protons in the nucleus

**Source Article:** ~/Effective_nuclear_charge

particle obeys Bose statistics. If the nitrogen nucleus had 21 particles, it should obey Fermi statistics, contrary to fact. Thus, Heitler and Herzberg concluded: "the electron in the nucleus

**Source Article:** ~/Discovery_of_the_neutron

. . .

be difficult to detect by available techniques.

About the time of Rutherford's lecture, other publications appeared with similar suggestions of a proton–electron composite in the nucleus

of cells in the center and most medial portion of the brain stem.

In order from caudal to rostral, the raphe nuclei are known as the nucleus

its conformational structure to expose the phosphorylation motif. The protein can be found in the cytoplasm and the nucleus, however most of the active complexes are found in the nucleus

. . .

way, acts as a gate creating a more depolarized state within the accumbens neurons although this depolarized state is much more transient. All in all, nucleus

then grows by accretion, as additional vapour molecules happen to hit it. The Kelvin equation, however, indicates that a tiny droplet like this nucleus

. The residence, planned as the sojourn for his sons Milan and Mihailo, was finished in 1836 and included a large garden, nucleus

The proton is the smallest particle of an atom. It is a positively charged particle. It is found in the nucleus of an atom

Though useful as a predictive model, the resulting screening constants contain little chemical insight as a qualitative model of atomic structure.

Comparison with nuclear charge

Nuclear charge is the electric charge of a nucleus of an atom

**Source Article:** ∼/Effective_nuclear_charge

separate them. Therefore, the closer quarks are to each other, the less the strong interaction (or color charge) is between them; when quarks are in extreme proximity, the nuclear force between them is so weak that they behave almost as free particles. This is the reason why the nucleus of an atom

**Source Article:** ∼/David_Gross

Orbital Angular Momentum

When quantum mechanics refers to the orbital angular momentum of an electron, it is generally referring to the spatial wave equation that represents the electron's motion around the nucleus of an atom

**Source Article:** ∼/Orbital_motion_(quantum)

. . .

allows us to make the substitution

where and are the energies of the two states at time , given by the diagonal elements of the Hamiltonian matrix, and is a constant. For the case of an atom

chain transfer and branching steps considered next.

Chain transfer

In some chain-growth polymerizations there is also a chain transfer step, in which the growing polymer chain RMn° takes an atom

College in Perth, graduating with a BSc in 1956. This was followed by a master's degree at Australia's first cyclotron, where he began his work as a high-energy physicist. His thesis from the University of Melbourne was on Coulomb excitations of the atom

. . .

small forest clearings in recent years.

The nominate subspecies P. r. rapae is found in Europe, while Asian populations are placed in the subspecies P. r. crucivora. Other subspecies include atom

Middle Irish popular etymology, but it is more likely a term for a "spell of concealment". It is also known by its incipit (repeated at the beginning of the first five sections) atom

perico Rhynchosia tarphanthaSchrankia
Schrankia distachya – sierillaSenna
Senna alata Senna atom

---

Q: What is the difference between diffraction and refraction?

A: Refraction is the scattering

Royal Navy ship names Atmospheric diffraction is manifested in the following principal ways:

Optical atmospheric diffraction

Radio wave diffraction is the scattering

**Source Article:** ∼/Atmospheric_diffraction

arrays A sol is a colloidal suspension made out of tiny solid particles in a continuous liquid medium. Sols are stable and exhibit the Tyndall effect, which is the scattering

**Source Article:** ∼/Colloid

s rule can also be stated in terms of the scattering time:

where $\tau$ is the true average scattering time and $\tau$impurities is the scattering

**Source Article:** ∼/Electron_mobility

. . .

is the ratio of scattering efficiency to total light extinction (which includes also absorption), for small-particle scattering of light. That is, , where is the scattering

as (we assume there is a finite ranged scatterer at the origin and there is an incoming plane wave along the -axis):

where is the scattering

mechanics describes low-energy scattering. For potentials that decay faster than as , it is defined as the following low-energy limit:

where is the scattering

. . .

helilifted to a defensive position at Ban Na. On 10 August 1969, PAVN troops overran Xieng Dat, scattering

evacuated by ship. The fallout of the Revolution resulted in an estimated 75,000 to 100,000 Black Americans scattering

with the attack on Phung Duc the PAVN 24th Regiment supported by tanks attacked the ARVN 45th Regiment on Hill 581 scattering

Q: What is the difference between diffraction and refraction?

A: Refraction is the scattering of electromagnetic

---

also called Lorenz-Mie theory or Lorenz-Mie-Debye theory, is a complete analytical solution of Maxwell's equations for the scattering of electromagnetic

**Source Article:** ∼/Light_scattering_by_particles

---

bomber.

In the 1960s Ufimtsev began developing a high frequency asymptotic theory for predicting the scattering of electromagnetic

**Source Article:** ∼/Pyotr_Ufimtsev

---

, Mathieu functions can be used to describe a variety of wave phenomena. For instance, in computational electromagnetics they can be used to analyze the scattering of electromagnetic

**Source Article:** ∼/Mathieu_function

---

. . .

---

Polaritonics is an intermediate regime between photonics and sub-microwave electronics (see Fig. 1). In this regime, signals are carried by an admixture of electromagnetic

---

in a single direction at a time.

Sonar also uses beamforming to compensate for the significant problem of the slower propagation speed of sound as compared to that of electromagnetic

---

and electrons are only concavities in the aether.

Mass and speed

Thomson and Searle
Thomson (1893) noticed that electromagnetic

---

. . .

---

Fictional characters involved in incest
Female characters in television
Teenage characters in television
Branning family A pinch (or: Bennett pinch (after Willard Harrison Bennett), electromagnetic

---

Hunters"

Season 6 (2010)

"Camp Leatherneck"
"DEA Takedown"
"Electronic Armageddon", electromagnetic

---

the particle may be then measured by the intensity of scintillation light produced by the various scintillator slices. An example detector that uses a shashlik electromagnetic

Q: What is the difference between diffraction and refraction?

A: Refraction is the scattering of electromagnetic waves

---

is based on the theories of Planck and the interpretation of those theories by Einstein. The correspondence principle then allows the identification of momentum and angular momentum (called spin), as well as energy, with the photon.

Polarization of classical electromagnetic waves

**Source Article:** ∼/Photon_polarization

---

Spin angular momentum may refer to:

Spin angular momentum of light, a property of electromagnetic waves

**Source Article:**
∼/Spin_angular_momentum_(disambiguation)

---

x 10-7 H/m.
= relative magnetic permeability of the material. Magnetically conductive materials such as copper typically have a near 1.

This velocity is the speed with which electromagnetic waves

**Source Article:** ∼/Speed_of_electricity

---

. . .

---

longitudinally, and "time-like" polarized waves in the four-potential. The transverse polarizations correspond to classical radiation, i.e., transversely polarized waves

---

a little thick just to explain the schematic although in reality, the air gap is so thin between prism and metal layer.

Waveguide coupling

The electromagnetic waves

---

pulses. This change in phase is caused by the difference in the number of wave cycles (or wavelengths) along the propagation path for horizontal and vertically polarized waves

---

. . .

---

Proton includes substantially more modules, including two looping ASR envelopes, two LFOs (one more than the Neutron), another filter, a waves

---

Music Easel (pictured), use a different method of timbre generation from Moog synthesizers. Moog units use oscillators with basic function generator type waves

---

SYNC" controllers are reminiscent of the Poly-Mod feature in the Sequential Circuits Prophet-5.
The microKORG XL also includes a waves

