# OpenReview forum: "Global-to-Local Support Spectrums for Model Explainability"
_ICLR.cc/2025/Conference — Submitted to ICLR 2025_

### Official Review · Reviewer_4Zwk · 2024-10-23

**Soundness:** 1
**Presentation:** 3
**Contribution:** 1
**Rating:** 1
**Confidence:** 4

**Summary:**

The paper proposes a novel method to retrieve points that serve to explain a classification of a point x_t in terms of training data points x_i. In particular, instead of selecting a single reference point x_i, a 'spectrum' of points is constructed which support the classification decision (relative to another class). The points in the spectrum are sorted according to their inner product (in feature space) with x_i from highest to lowest inner product. The spectrum ends at the last poiint that is still in the same class as x_i. The method is tested on some examples from MNIST and RedPajama / Wikipedia subset.

**Strengths:**

1. The paper correctly identifies the problem that many current methods to select reference points from the training data that support classification are not always convincing and tend to be too global.
2. The proposed method is relatively simple to understand and well-motivated/derived from the theory of linear classifiers.
3. The presentation of the method is clear and nicely illustrated with figures.
4. The discussion of samples in the experiments is insightful and reveals some clever uses of the proposed method.

**Weaknesses:**

1. The simplicity of the proposed method also limits its delta beyond prior work. In one sentence, the proposed method could be summarized as: consider points descendingly according to their inner product with the test point and, among those, take the one with lowest prediction score for the test points' class. This does not strike me as a substantial step beyond existing approaches.
2. The paper appears empirically extremely weak. The evaluation is only anecdotally performed on a few examples. For the text case in particular, the proposed method seems to be a rather unnatural fit given the adjustments needed to make the method applicable (removing relative scores and considering next-word prediction as a classification problem). The anecdotal results in the second case also seem only partially convincing. At least one example that reveals generation close to an example from the training set would have been nice to indicate that the method is useful for the respective example case. Generally speaking, a user study would have been much superior to provide a convincing evaluation.
3. In addition to the lack of proper evaluation for the proposed method, a comparison to baseline methods is entirely missing. This, by itself, appears as a fatal flaw to me.
4. The paper hardly cites any prior work, except in section 4. This appears scholarly unacceptable, especially in the introduction, which is full of claims that ought to be backed up by citations. Further, the bibliography appears severely limited, ignoring most of the recent literature on explainability and interpretability (refer, e.g., to the review of Madsen et al., https://doi.org/10.1145/3546577 ).

Overall, it seems to me that this paper, unfortunately, fails to comply basic research practices of the field and thus is far removed from meeting the acceptance threshold.

**Questions:**

1. If I mischaracterized the method, I am happy to be corrected. Could this be checked?
2. Why was the current evaluation design selected? Why wasn't a user study considered, instead?
3. Why was no baseline studied to compare against?
4. Why are there no citations in the introduction?

---

### Official Review · Reviewer_wAGW · 2024-10-28

**Soundness:** 2
**Presentation:** 3
**Contribution:** 3
**Rating:** 5
**Confidence:** 3

**Summary:**

The paper introduces a novel approach for generating model explanations of deep models in classification settings by constructing “Support Spectrums.” The method utilizes the concept of a support set, a set of training points that lie closer to the decision boundary than the test sample of interest. The authors then showcase how existing sample importance measures can be decomposed into global and local importance measures. These are then combined with the support set into a parametrized set called the global-to-local support spectrum. The authors showcase the method through examples on MNIST image classification and text generation tasks using large language models on the Wikipedia data subset.

**Strengths:**

- The derivation of the method seems technically correct, and the proposed method appears novel.
- The support spectrum of a point seems to be a good proxy for how hard a sample is for the model / how likely the sample was mislabeled.
- The qualitative examples look interesting and are presented well.
- The authors show how the method can be used to debug a dataset for text generation, specifically investigating a potential case of spurious correlation.
- The code was reproducible on my machine with some minor tweaks.

**Weaknesses:**

The authors criticize existing sample-based methods, stating that they tend to be skewed by outliers, such as mislabeled points or points that are very close to the decision boundaries. This claim is not well supported; the shortcomings of existing approaches are only demonstrated on a toy example and does not seem to be a strong basis for criticizing current approaches.

The authors then use the approaches they criticize in the derivation of their method. Demonstrating how the proposed method overcomes the shortcomings of those approaches empirically would significantly help the justification of the method.

The authors do not examine the computational costs of generating a support spectrum, which seems to be non-trivial for deep models.

**Questions:**

- L231: “Empirical results show that using Wc − Wk and bc −bk instead of Wc and bc yields a better spectrum relative to k.” Where is this demonstrated? This could be incorporated into the paper or added to the Appendix.
- What do the authors mean by path on L158?
- The size of the support spectrum of a point seems to be a good proxy for how hard a sample is for the model / how likely the sample was mislabeled. It would be interesting to compare the approach to existing methods for finding hard / easy samples, such as [1], or identifying mislabeled samples [2].
- I would suggest moving the related works section after the introduction. Influence function and representer point methods are introduced in the abstract and introduction. They are then used in the derivation of the proposed method (2.3, 2.4) and afterwards stated again in the related works section (4.).
- I would also consider expanding on how the proposed method compares to the existing methods.
- L192: (b) the green class?
- L158: “show the general spectrums (as points along the paths)" it is not clear what the paths are here (varying the $\gamma$ parameter?). This is the only time a path is mentioned in the paper (or source code)
- L288: Would “row corresponding to k=5” be slightly more clear than "row 6”?
- L314: TD-IDF is used without stating Term Frequency-Inverse Document Frequency
- L511: the global to local support spectrums

[1] Swayamdipta, Swabha, et al. "Dataset Cartography: Mapping and Diagnosing Datasets with Training Dynamics." Proceedings of the 2020 Conference on Empirical Methods in Natural Language Processing (EMNLP). 2020.

[2] Northcutt, Curtis, et al. "Confident learning: Estimating uncertainty in dataset labels." Journal of Artificial Intelligence Research 70 (2021): 1373-1411.

---

### Official Review · Reviewer_Wyz6 · 2024-11-03

**Soundness:** 2
**Presentation:** 2
**Contribution:** 2
**Rating:** 3
**Confidence:** 4

**Summary:**

The paper proposes a method to generate a support set for a prediction from a model, that demonstrate how well the test point can be distinguished from the points not in the predicted class.

**Strengths:**

- interesting problem
- evaluation on both image and text data

**Weaknesses:**

- there are no baselines for comparison and no quatitative or qualilative evaluation, only a few case studies
- limited related work
- the support set is similar to counterfactual explanations, with the difference being counterfactual explanantions are in the opposite class. But why is support set better than counterfactual explanations? Or is it just an alternative?

**Questions:**

the support set serves s similar purpose than counterfactual explananations. why not just use counterfactual explanation? Why do the examples need to be in the predicted class?

---

### Official Review · Reviewer_eaG1 · 2024-11-04

**Soundness:** 2
**Presentation:** 1
**Contribution:** 2
**Rating:** 3
**Confidence:** 3

**Summary:**

The paper introduces a novel method for providing explanations for model predictions using "support spectrums." This method aims to overcome limitations in existing sample-based methods such as influence functions and representer points, which are often biased towards outliers and lack specificity for individual test points.

**Strengths:**

The authors introduce a new type of explanation called a "support spectrum," which provides a transition from global to local importance for individual test points. The support spectrum includes training points that "lie in between" the test point and the points in other classes, offering tailored explanations for each test case.

**Weaknesses:**

Major:

1) One of my main concern with the paper is that the paper is difficult to follow; some explanations of the paper are difficult to comprehend; for instance (See abstract & introduction), the authors claim that their method can be used for explaining a specific test sample, but influence functions would do the same; or, they refer to outlier (introduction paragraphs 3) as "mislabeled point" or "points close to the decision boundary", which are not accurate. I am afraid such inaccurate statements is prevalent in the paper.

2) Most importantly, I am not sure if the paper contributes significantly to the existing methods (influence function and representer points). First of all, the spectrum set obtained from optimization problem (1) with constraints (2) and (3) does not seem to guarantee the non-existence of "outliers" - as far as I could understand from the paper, it is less likely, but not impossible; if we get the spectrum sets with "outliers", then there is no use in the constructing the set for better explainability.

On the other hand, if we develop a simple procedure to compute the "outliers" (e.g., the mislabeled points and the points near the classification boundary are not difficult to obtain indeed), remove those points from the set, and apply other explainable methods (like influence function or representer points), then all the drawbacks mentioned in the paper are covered and there is no need for developing the method. I therefore think that the paper has little contribution to offer.

**Questions:**

The authors mention using the approach in text generation tasks by treating the generation as a sequence of autoregressive classification tasks. However, generating support spectrums for each token can become impractical for longer text sequences or large vocabulary sizes, as computing the relative importance of many training samples for each generated token can significantly increase computational requirements. Can authors comment on this point and how they handled it in their experiments?

---

### Meta-Review · Area_Chair_QwHL · 2024-12-11

**Metareview:**

This paper proposes support spectrums for interpretability, with the aim of improving over sample-based methods. The method is evaluated on two modalities (text and images) and support spectrums might provide an interesting way to find difficult examples. A case study is also presented to identify spurious correlation in text data.

Major flaws were reported by the reviewers, including the lack of proper quantitative evaluation and comparison to baselines as well as limited contribution and references. These concerns were not addressed by the authors.

Based on the reviews and scores, I believe there are fundamental pieces missing in this paper that prevent publication at this time.

**Additional Comments On Reviewer Discussion:**

There was no further discussion as the reviewers all suggested rejection and the authors did not engage in the discussion. In a private discussion, I questioned the strong score of 1 but it was maintained and justified by the reviewer and seemed reasonable to me after their arguments.

---

### Decision · Program_Chairs · 2025-01-22

Reject